complexity/computational biology/evolution

linguistic niche hypothesis, language complexity, typology, non-native speakers, entropy

**Author for correspondence:**
Alexander Koplenig
e-mail: koplenig@ids-mannheim.de

# Language structure is influenced by the number of speakers but seemingly not by the proportion of non-native speakers

## Alexander Koplenig

Institute for the German Language (IDS), Mannheim, Germany

 AK, 0000-0002-9630-9680

Large-scale empirical evidence indicates a fascinating statistical relationship between the estimated number of language users and its linguistic and statistical structure. In this context, the linguistic niche hypothesis argues that this relationship reflects a negative selection against morphological paradigms that are hard to learn for adults, because languages with a large number of speakers are assumed to be typically spoken and learned by greater proportions of adults. In this paper, this conjecture is tested empirically for more than 2000 languages. The results question the idea of the impact of non-native speakers on the grammatical and statistical structure of languages, as it is demonstrated that the relative proportion of non-native speakers does not significantly correlate with either morphological or information-theoretic complexity. While it thus seems that large numbers of adult learners/speakers do not affect the (grammatical or statistical) structure of a language, the results suggest that there is indeed a relationship between the number of speakers and (especially) information-theoretic complexity, i.e. entropy rates. A potential explanation for the observed relationship is discussed.

## 1. Introduction

In an influential and widely cited paper, Lupyan & Dale [1] present striking large-scale evidence for a statistical relationship between the estimated number of language users and structural properties of languages, especially several factors related to morphological complexity, e.g. the use of inflections to mark grammatical relationships in a sentence. The paper demonstrates that languages with more speakers tend to have simpler

inflectional morphology compared to languages spoken by smaller groups: on an overall index of morphological complexity, languages with fewer speakers tend to score higher than languages with many speakers [1]. As inflections on nouns often tend to change the form of modifiers, verbs or adjectives (a requirement of correspondence known as agreement in linguistics), increased morphological complexity leads to an increased redundancy in language [2]. Lupyan & Dale [1] equate this kind of linguistic redundancy to information-theoretic redundancy, i.e. to what extent a given text can be compressed [3]. Lupyan & Dale [1] demonstrate that, for translations of a standard text (the Universal Declaration of Human Rights) into more than 100 different languages, languages with more speakers tend to be less informationally redundant. Interestingly, from an information-theoretic point of view, data compression, i.e. the entropy rate of a source, can also be interpreted as a measure of complexity [4]: the smaller the degree of redundancy in a string, the harder it is to predict subsequent text based on previous input [5], and the greater its complexity [6]. Thus, the results of [1] indicate that languages with more speakers are morphologically *less* complex, but informationally *more* complex. This points towards a negative statistical association between morphological complexity and entropy rates.

To explain their results, the authors argue (in [1] and in subsequent papers [7–9]), in analogy to biological organisms adapting to their ecological niche, that this relationship can be best understood as resulting from languages adapting to the social environments in which the languages are spoken and learned. In this context, the linguistic niche hypothesis assumes that languages that are spoken by more people over greater geographical areas will, on average, also be learned by a greater proportion of non-native (henceforth: *L2*) learners, i.e. often adults. As complex morphology appears to be difficult to learn for adults [7,10–12], the linguistic niche hypothesis conjectures that there should be a negative selection over time against such hard-to-learn morphological paradigms for languages with a larger number of *L2* speakers compared to languages that are mainly learned during childhood as a native language, i.e. by children (henceforth: *L1*). This, in turn, then explains the observed negative statistical association between speaker population size and morphological complexity.

The idea of a potential influence of the proportion of *L2* speakers serves as an important point of reference in studying and understanding various aspects of the structure and origin of natural languages, e.g. [13–23]. However, it is important to point out that because the argument outlined above is inductive by nature, its validity cannot be simply taken (more or less implicitly) for granted. Crucially, Lupyan & Dale [1] use the estimated speaker population size as a proxy for the proportion of *L2* speakers [2]. It is an empirical question, whether this is appropriate because there are also potential other mechanisms that could explain the relationship between social and linguistic structure without referring to the proportion of *L2* speakers [2,9,24–26]. In the example of morphological case, [27] present evidence that the proportion of *L2* speakers is indeed statistically associated with morphological (case) complexity. However, it is not clear if the sample of [27] is unbiased, because (i) it only comprises 66 languages; (ii) compared to the median estimated speaker population size of 7000 for the roughly 7000 languages listed by the *Ethnologue* [1], the median estimated speaker population size in the sample of [27] is 14 200 000; (iii) all 66 languages have an estimated proportion of *L2* speakers that is greater than zero with a median estimate of roughly 25% which, based on the assumption of the linguistic niche hypothesis that most languages have almost no *L2* speakers [1,9], is rather high; and (iv) there is no (Spearman) correlation between the estimated speaker population size and the estimated proportion of *L2* speakers ($r = 0.060$). As mentioned above, this is actually a key assumption of the linguistic niche hypothesis. Interestingly, [28, p. 133] also notes that there is no correlation between population size and the *L2* proportion for a similar dataset that consists of 110 languages and that is used to test the influence of the proportion of *L2* speakers. Both [29] and [17] do not find clear evidence for a relationship between morphological complexity and the proportion of *L2* speakers.

Therefore, a systematic test of the relationship between speaker population structure and the linguistic and statistical structure of human languages with as many languages as possible is important, given its relevance for linguistics and, more generally, cultural evolution theory and anthropology. Or, as [2, p. 1832] puts it: 'the most intriguing follow-up question is what mechanism could cause these patterns to emerge. Understanding the link between the microscale of individual behaviour (speech in this case), and the macro-scale of historically enduring shared patterns of culture (the grammars of languages in this case), is the most challenging issue in the study of cultural evolution, and indeed in anthropology more generally.'

Up until recently, however, as [27] point out, estimations regarding a breakdown of *L1* versus *L2* populations were very limited. In its 19th edition, the *Ethnologue* now includes data on *L2* users where they are known [30]. In addition, the *Ethnologue* categorizes each language in regard to how

endangered it is using the expanded graded intergenerational disruption scale (henceforth: *EGIDS*) [31]. In this context, a language is categorized as *vehicular*, if it is used as an *L2* in addition to being used as an *L1*. This information can be used to indirectly gain information about the proportion of *L2* users: 'A language at EGIDS 4 or lower is, by definition, a local language and L2 users are not expected. However, languages at EGIDS 3 and higher are vehicular and, by definition, they should have a significant number of L2 users' [32]. The great advantage here is that information of the *EGIDS* level is available for all languages that are listed in the *Ethnologue*.

In this paper, I use available information on the number of *L2* users and on *vehicularity* as an indicator for whether a language is used by *L2* speakers, to test the assumed relationship between the proportion of adult speakers and morphological and information-theoretic complexity. If the linguistic niche hypothesis holds, we should expect a high proportion of *L2* users to be statistically associated with low morphological complexity and high entropy when statistically controlling for the speaker population size.

If, on the other side, there is a link between language structure and speaker population size that is independent of the proportion of *L2* speakers/*vehicularity*, then this would indicate that a theory that aims to explain the emergence of this association needs to review (and test) other possible mechanisms that might be relevant in this context.

# 2. Material and methods

## 2.1. Data

The data used in this paper are from eight different sources. Basic information on different languages and genealogical classifications are taken from [33]. Speaker population estimates and geographical range size estimates are taken from [34]. Information on geographical language areas are taken from [35]. Data on morphological complexity are taken from [36]. The entropy rates are based on estimations for parallel Bible translations published by Koplenig *et al.* [37]. Word entropy estimates as a measure of lexical diversity (cf. electronic supplementary material, §2) are taken from [38]. Aggregated *L2* speaker proportions are taken from [39] and from [40]. The different sources are merged via the three letter language-specific ISO 639-3 code.

## 2.2. Population estimates

Speaker population size and geographical range size estimates are taken from [34], who report the total number of *L1* speakers based on information from [39] and calculate range sizes in square kilometres based on information from [41]. Aggregated information on *vehicularity* and *L2* proportions are taken from [39]. Languages with an *EGIDS* value of 1, 2 or 3 are categorized as *vehicular*, while languages with an *EGIDS* value of 4–10 are categorized as *non-vehicular*. The proportion of *L2* speakers is calculated as follows:

$$p_{L2} = 1 - \frac{N_{L1}}{N}, \tag{2.1}$$

where $N_{L1}$ is the estimated number of *L1* users, and $N$ is the estimated total number of all users. In correspondence with the categorization scheme of the *Ethnologue* [32], *non-vehicular* languages with no available information on *L2* users are assigned an *L2* proportion of 0. However, while the *Ethnologue* states that for *non-vehicular* languages *L2* users are not expected, there are in total 78 *non-vehicular* languages for which the *Ethnologue* reports an *L2* proportion greater than 0 (with a median estimate of 0.086). To rule out the possibility that those exceptions to the rule in the *Ethnologue* categorization scheme affect the results, separate analyses in which those languages are dropped are presented in electronic supplementary material, §7.

Additional analyses where (i) *N*, i.e. the total number of speakers is used instead of the number of *L1* speakers and where (ii) only languages that are categorized as *vehicular* are being included are presented in electronic supplementary material, §§8 and 10.

## 2.3. Morphological complexity

To construct an index of morphological complexity, [36] information is extracted on 28 relevant features of morphology from the World Atlas of Languages Structures [42] (henceforth: *WALS*). For example, the

WALS ch. 30A 'Number of Genders' gives a range of 5 values from 'None' to 'Five or more'. Those values are then mapped to the values 1–5, where higher values are indicative of higher complexity. The values of each feature are normalized to the interval [0,1]. The morphological complexity score $C$ is then calculated by summing the normalized features divided by the number of available features. Let $f_i$ be the normalized value of feature $i$ and $N_F$ be the number of features that are available in the corresponding languages, then $C$ can be written as follows:

$$C = \frac{1}{N_F} * \sum_{i=1}^{N_F} f_i. \tag{2.2}$$

Greater values are indicative of more morphological complexity. For more details and a list of all used WALS features, cf. [36]. In total, there are 1713 languages with at least one available feature. It is important to note that the amount of available WALS information varies greatly for different languages [1,2], e.g. there are only 10 languages for which information on all 28 features is available [36], but there are 393 understudied languages with only one or two available features. To account for this data sparseness, I present separate analyses on the full dataset (at least one available feature) and on a subset of languages where at least six features are available (50% of all languages have information on at least six features).

## 2.4. Information-theoretic complexity

To measure the redundancy of a symbolic sequence, i.e. a book, string or a text, it is possible to apply one of the key ideas of the minimum description length principle: 'any regularity in the data can be used to compress the data, i.e. to describe it using fewer symbols than needed to describe the data literally. The more regularities there are, the more the data can be compressed' [3]. Or put differently, the higher the degree of redundancy in a text, the easier it is to predict the next character in a text after having read the preceding text. From an information-theoretical point of view, the redundancy of a given string can be measured by estimating the entropy per symbol that can be considered the 'ultimate compression' of the string [43], i.e. the smallest number of bits (normalized by the length of the string) required to construct a compressed version of the text, given that the original string can be perfectly reconstructed from its compressed representation. Here, I use estimates for the Gospel of Mark in more than 1000 different languages based on the Parallel Bible Corpus [33] that are taken from [37]. For languages with more than one available translation, entropy estimates are averaged.

Entropy rates are estimated on the basis of the non-parametric method of [4,44] that builds on the key idea of the Lempel–Ziv compression algorithm [45]. This method does not require any prior training, produces robust estimates without the need for very long strings as input and is able to take into account the very long-range correlations typical of literary texts [46,47] that are not captured by direct parametric Markovian or 'plug-in' estimators [4]. If we represent a text $t$ as a symbolic sequence of $N$ characters, i.e. $t = \{c_1, c_2, \ldots, c_{N-1}, c_N\}$ where $c_i$ represents any character (including white spaces and punctuation marks) in the text at position $i$. The entropy rate can be estimated as [4, cf. eq. (2.1)]:

$$H_t = \left[ \frac{1}{N} \sum_{i=2}^{N} \frac{l_i}{\log(i)} \right]^{-1}. \tag{2.3}$$

To measure the minimum number in bits per character [bpc], logarithms are taken to base two. Here, the key quantity of interest is the match-length $l_i$. To determine the redundancy at position $i$, we examine the whole portion of the text up to (but not including) $i$ and monitor how many of the initial characters of the text portion starting at $i$ have already occurred in the same order somewhere in the preceding text, and record the length of longest continuous substring. Our key quantity of interest $l_i$ is obtained by adding 1 to the longest match-length. There are no restrictions regarding the size of the 'database', illustrating (i) why the estimator can be used in the presence of very long-range correlations, as we do not impose any restrictions on how far 'into the past we can look for a long match' [4] and (ii) that the estimator seems like a reasonable model of linguistic patterns of experience, as it captures structure at various levels of linguistic organization (co-occurring words, regular relations between grammatical word forms, constructions) that can be linked to theories of language learning and language processing [48]. More details of the approach can be found in [37].

In total, information for 2143 languages could be obtained (1088 data points for entropy rates, 1581 data points for morphological complexity). Of those languages, 1902 are categorized as non-vehicular,

while the remaining 241 languages are *vehicular*. The median estimated speaker population size is 15 000; the median estimated proportion of *L2* speakers is 0 for *non-vehicular* languages and 0.233 for *vehicular* languages.

## 2.5. Statistical analysis

According to the definition of the *Ethnologue*, a *vehicular* language should have a significant number of *L2* users, while no *L2* users are expected for *non-vehicular* languages. To test whether the argument of the linguistic niche hypothesis is correct, *vehicularity* should be a significant predictor of morphological and information-theoretic complexity after controlling for the influence of estimated speaker population size. In what follows, I use a generic version of Still and White's permutation test [49]. First, complexity (morphological/information-theoretic) is regressed onto the log of population size and residuals are obtained. Secondly, the residuals are regressed on the one/zero variable *vehicularity* and the proportion of explained variance, denoted as $R^2$, is calculated. Because the influence of the speaker population size has been regressed out, the resulting $R^2$ can then only be attributed to the influence of *vehicularity* as a proxy of the proportion of adult speakers, but not to the speaker population size. To test the statistical significance of this relationship, the residuals are randomly permuted 100 000 times. Let $c$ denote the number of times where the $R^2$ of the derived dataset is *greater than or equal to* the value of the $R^2$ computed on the original data. A corresponding coefficient is labelled as 'statistically significant' if $c < 1000$, i.e. $p < 0.01$. Note that this *p*-value is equal to the *p*-value of *t*-statistic of the actual *β*-coefficient of *vehicularity*. In addition to the actual *p*-values, Bonferroni-adjusted *p*-values are presented to correct for multiple testing [50]. Let $m$ denote the number of conducted tests; then the adjusted critical significance level is the chosen significance level of 0.01 divided by $m$. For example, if six tests are being conducted, a critical approximate level of $0.01/6 \approx 0.0017$ is adopted.

In addition to the model with only the log of the speaker population size as a fixed effect, the first step of the test is repeated for models that include random intercepts and slopes that can vary across linguistic families and geographical areas to control for the non-independence of data points due to genetic and areal relationships between languages [51]. Estimates are derived by maximum likelihood.

In addition, electronic supplementary material, §1 presents the results of linear mixed effects models, where morphological and information-theoretic complexities are predicted by (i) fixed effects of *vehicularity*, the logged speaker population size, and their interaction and (ii) various random effects for language families and geographical areas. A validation of the permutation test is presented in electronic supplementary material, §5. Two further tests are presented in electronic supplementary material, §§6 and 9: a different permutation test approach [52] and a binary mediation analysis that tests whether the link between population size and complexity is mediated by the proportion of *L2* speakers.

## 2.6. Correlation analysis

To understand the relationship between two variables *v1* and *v2* without making any assumptions regarding the functional form of the relationship, I use the non-parametric Spearman correlation coefficient denoted as $r_{v1v2}$. It assesses whether there is a monotonic relationship between two variables and is computed as Pearson's correlation coefficient on the ranks and average ranks of *v1* and *v2*. Spearman part correlations (also called semi-partial correlations) for *v1* and *v2* after removing the effect (partialling out) of a third variable *z* from *v2*, are obtained by fitting linear regressions of the ranks of *v2* on the ranks of *z* and obtaining residuals denoted as *e2*. The part Spearman correlation $pr_{v1v2z}$ is then calculated as the Pearson correlation between *e2* and the ranks of *v1*. Alternatively and illustratively, it can be calculated as follows:

$$pr_{v1v2z} = \frac{r_{v1v2} - r_{v1z} \times r_{v2z}}{\sqrt{1 - r_{v2z}^2}}. \tag{2.4}$$

Significance of the observed coefficient for *v1* and *v2* (and *z*) is determined by Monte Carlo permutation tests where the observed values of *v1* are randomly permuted 100 000 times to conserve the covariance pattern of *v2* and *z* as suggested by Manly [53]. As for the permutation test above, statistical significance is assessed by counting the number of times the value of the correlation coefficient computed on the randomly permuted dataset is (i) *greater than or equal to* the value of the *positive* correlation coefficient

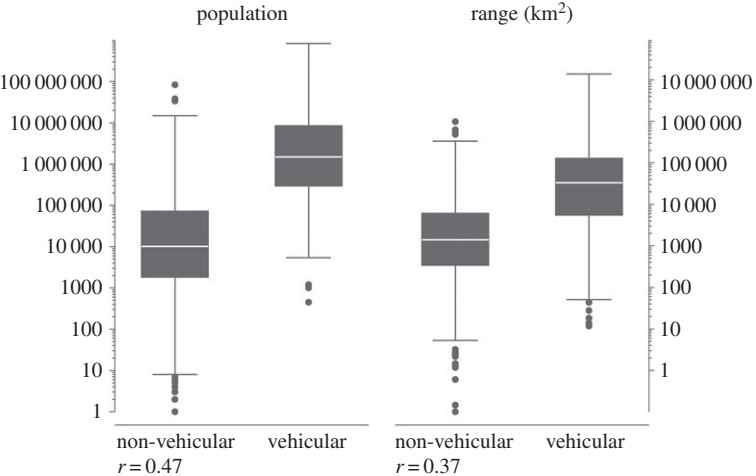

**Figure 1.** Box plots of the speaker population size and the geographical range size for non-vehicular ($N = 1902$) and vehicular ($N = 241$) languages.

computed on the original data or (ii) *less than or equal to* the value of the *negative* correlation coefficient computed on the original data.

To account for statistical non-independence within and between language families and areas, the electronic supplementary material contains two further analyses (electronic supplementary material, §§3 and 4). Firstly, Monte Carlo simulations with 100 000 repetitions are conducted. To avoid a disproportionate influence of individual cases, families/areas with less than five members are aggregated into an 'other' group. Per repetition, one observation is randomly drawn from each language family/area. Based on this random sample, Spearman correlations and Spearman part correlations are calculated as described above. The values presented below denote $z$-transformed average values over repetitions, i.e. the simulated correlation values are first transformed using the inverse hyperbolic tangent function. Then the average of the transformed values is calculated and back-transformed to a correlation value using the hyperbolic tangent function. Secondly, separate analyses are conducted for six language families and for six language areas.

## 3. Results

Figure 1 demonstrates that *vehicularity* correlates positively with the logged speaker populations size ($r = 0.474$) and logged geographical range size ($r = 0.373$). In accordance with the linguistic niche hypothesis, this indicates that with 'increased geographical spread and an increasing speaker population, a language is more likely to be subjected to learnability biases and limitations of adult learners' [1, p. 7]. Thus, *vehicularity* can be used to indirectly test the inductive argument of the linguistic niche hypothesis.

Table 1 presents the results of the permutation tests. The results demonstrate that *vehicularity* only significantly predicts morphological/information-theoretic complexity in a model without control for potential confounding variables. In all models with fixed control for the estimated speakers population size (logged) and random controls for language families and areas, the coefficient of determination for *vehicularity* is below 1% and does not achieve significance (at $p < 0.01$). This result questions the idea that large proportions of *L2* speakers affect the morphological and statistical structure of languages.

The results of the validation of the permutation test (see electronic supplementary material, §5) suggest that it is indeed population size that explains the apparent relationship between *vehicularity* and complexity (morphological/information-theoretic).

The results of the mixed effects models (see electronic supplementary material, §1) support the conclusion that *vehicularity* does not significantly predict morphological or information-theoretic complexity. Based on an analysis of 91 languages, [40] claim that languages with more *L2* speakers tend to have lower lexical diversities. In electronic supplementary material, §2, the results of the permutation tests with the unigram word entropy as measure of lexical diversity are presented for 1080 languages. Again, the results question the idea of a significant influence of large proportions of *L2* speakers.

**Table 1.** Results of the permutation test. 1st column: dependent variable. 2nd column: control variable (fixed). 3rd column: control variables (random). 4th column: percentage of explained variance. 5th column: direction of the relationship ('+' indicates a positive relationship, '−' indicates a negative one). 6th column: number of available languages. 7th column: number of included WALS features/chapters (if relevant). N.B.: for the 'no control' models, the dependent variable was directly regressed onto *vehicularity*, because this is equivalent to fitting constant-only models. The population size is logged in all models. Models with random slopes only include language families/geographical areas with at least 10 data points as suggested by Jaeger *et al.* [51]. Values are rounded for illustration purposes only. One asterisk (*) indicates that the corresponding coefficient passed the permutation test at $p < 0.01$. Two asterisks (**) denote statistical significance after the Bonferroni adjustment [$m = 21$].

| dependent variable | control variable (fixed) | control variable (random) | $R^2$ | direction | N | $N_F$ |
|---|---|---|---|---|---|---|
| morphological complexity | no control | | 1.38** | − | 1581 | 1 |
| | population size | | 0.27 | − | 1581 | |
| | | families | 0.24 | − | 1581 | |
| | | areas | 0.11 | − | 1512 | |
| | | families and areas | 0.11 | − | 1512 | |
| | | families (intercepts and slopes) | 0.35 | − | 1291 | |
| | | areas (intercepts and slopes) | 0.12 | − | 1512 | |
| | no control | | 1.92** | − | 862 | 6 |
| | population size | | 0.20 | − | 862 | |
| | | families | 0.18 | − | 862 | |
| | | areas | 0.16 | − | 821 | |
| | | families and areas | 0.15 | − | 821 | |
| | | families (intercepts and slopes) | 0.24 | − | 654 | |
| | | areas (intercepts and slopes) | 0.13 | − | 809 | |
| entropy rate | no control | | 14.68** | + | 1088 | |
| | population size | | 0.28 | + | 1088 | |
| | | families | 0.03 | + | 1088 | |
| | | areas | 0.01 | − | 719 | |
| | | families and areas | 0.00 | − | 719 | |
| | | families (intercepts and slopes) | 0.00 | − | 912 | |
| | | areas (intercepts and slopes) | 0.03 | − | 695 | |

Both the alternative permutation test approach (see electronic supplementary material, §6) and the binary mediation analysis (see electronic supplementary material, §9) support the results presented here.

## 3.1. Correlation analysis

Table 2 summarizes the results. As expected, row 1 of table 2 shows that there is a significant correlation between the estimated speaker population size and the proportion of *L2* speakers. Row 2 demonstrates that there is no significant negative correlation between morphological complexity and the *L2* proportion, either for $N_F \geq 1$, or for $N_F \geq 6$. Again, this questions the idea of an impact of *L2* speakers on the morphological structure of languages. Row 3 demonstrates that there is a weak but significant negative Spearman correlation between the morphological complexity index and the speaker population size that still passes the permutation test after partialling out the influence of the *L2* proportion. The entropy rate correlates significantly with the *L2* proportion (cf. row 4); however, when the effect of the speaker population size is removed, the resulting correlation coefficient is sharply reduced by a factor of roughly 3.5 to less than 0.09. In conjunction with the other results presented in this paper and as electronic supplementary material, the empirical evidence does not support a relationship between the two variables.

Row 5 reveals that there is a strong and significant positive correlation between the entropy rate and the speaker population size. From an information-theoretic point of view, this observation implies that languages with more speakers tend to be less redundant and therefore more complex. Row 6 shows

**Table 2.** Summary of the correlation analysis. 1st column: Row number (for reference in the main text). 2nd column: Spearman correlation coefficient between the specified variables. 3rd column: number of available cases to calculate $r_{v1v2}$. 4th column: Part Spearman correlation coefficient between the specified variables after controlling for the third specified variable. 5th column: number of available cases to calculate $pr_{v1v2z}$. 6th column: number of included *WALS* features/chapters (if relevant). N.B.: values are rounded for illustration purposes only. One asterisk (*) indicates that the corresponding coefficient passed the permutation test at $p < .01$. Two asterisks (**) denote statistical significance after the Bonferroni adjustment $[m = 15]$.

| row | $r_{v1v2}$ | | $N_r$ | $pr_{v1v2z}$ | $N_r$ | $N_F$ |
|---|---|---|---|---|---|---|
| 1 | $v_1$: speaker population size │ $v_2$: *L2* proportion | | | | | |
| | 0.259** | | 1991 | | | |
| 2 | $v_1$: morphological complexity │ $v_2$: *L2* proportion | | | $z$: speaker population size | | |
| | −0.044 | | 1450 | −0.014 | 1450 | 1 |
| | −0.066 | | 774 | −0.007 | 774 | 6 |
| 3 | $v_1$: morphological complexity │ $v_2$: speaker population size | | | $z$: *L2* proportion | | |
| | −0.122** | | 1581 | −0.101** | 1450 | 1 |
| | −0.164** | | 862 | −0.155** | 774 | 6 |
| 4 | $v_1$: entropy rate │ $v_2$: *L2* proportion | | | $z$: speaker population size | | |
| | 0.295** | | 986 | 0.089* | 986 | |
| 5 | $v_1$: entropy rate │ $v_2$: speaker population size | | | $z$: *L2* proportion | | |
| | 0.562** | | 1088 | 0.428** | 986 | |
| 6 | $v_1$: morphological complexity │ $v_2$: entropy rate | | | | | |
| | −0.003 | | 526 | | | 1 |
| | 0.057 | | 335 | | | 6 |

that there is no noteworthy monotonic relationship between the entropy rate and the morphological complexity index. This result questions the assumption of [1] that morphological specification and informational source-redundancy are correlated. The additional analyses (see electronic supplementary material, §§3, 4, 7, 8 and 10) generally support the results presented here.

## 4. Discussion

The results presented in this paper question the idea of an impact of non-native speakers on the grammatical and statistical structure of languages. *Vehicularity* does not correlate significantly with morphological and information-theoretic complexity when the effect of the estimated speaker population size is removed. It is, of course, important to ask if *vehicularity* is a good proxy for whether a language is used as an *L2*. Here is a corresponding interesting quote from the editors of the *Ethnologue* that shows how languages are classified in the *Ethnologue*: 'Based on the use of the phrase "vehicular language" by some as a synonym for lingua franca, we use the term vehicular to refer to the extent to which a language is used to facilitate communication among those who speak different first languages. *If a language is characterized here as being Vehicular, it is used by others as an L2 in addition to being used by the community of L1 speakers*' ([31]; my emphasis). Therefore, I believe that it is appropriate to use *vehicularity* to test the linguistic niche hypothesis. In a certain sense, using *vehicularity* as proxy is a direct quantitative test of the linguistic niche hypothesis that was suggested by [2, p. 1834/5]: 'One obvious hypothesis is that languages that have large populations but no adult learners (. . .) should look like small languages in terms of their complexity. The corollary is that languages that are small but for some exceptional reason spoken by a large proportion of adult learners should look grammatically like large languages. These hypotheses are clearly testable.' The presented results imply that when the effect of the speaker population size is controlled for, the (grammatical and statistical) structure of *vehicular* languages (i.e. languages with a significant number of *L2* users) does not seem to be different from that of *non-vehicular* languages (i.e. local languages

where *L2* users are not expected). In addition, the correlation analyses also question the idea of an impact of the relative proportion of *L2* speakers on the structure of languages. Thus, it seems that compared to other factors that are associated with speaker population size, adult *L2* learning and corresponding learnability limitations only seems to play a minor role (if any).

As an aggregated measure of morphological complexity was used in this paper, it is important to point out three things: (i) there are different ways of constructing an index of complexity as different studies show [1,36,54]. This highlights 'the need for a common analytical approach' as [54, p. 5] put it. (ii) Lupyan & Dale [1] present separate analyses for several different aspects of morphological complexity, regarding quantitative grammatical measures and qualitative grammatical types. It would be important to test the predictions of the linguistic niche hypothesis regarding the influence of the relative proportion of *L2* learners on such a fine-grained level, e.g. for different grammatical domains, too. (iii) Further (exploratory) research that critically examines the suitability of different grammatical features and the corresponding (e.g. *WALS*) coding principles for studies that quantify complexity to test hypotheses on linguistic adaption is clearly important as [29] demonstrate.

In addition, there is no noteworthy negative correlation between morphological complexity and entropy rates. This result challenges the idea that linguistic redundancy that arises from morphological specification covaries with information-theoretic redundancy as assumed by [1,2,8]. The reason for this result might be that intra-word redundancy is traded off against inter-word redundancy, i.e. word order. For example, [37] present large-scale evidence for an inverse relation between the amount of redundancy contributed by the ordering of words and the amount of redundancy contributed by the internal word structure: languages that rely more on word order to transmit grammatical information, rely less on intra-lexical regularities and vice versa. Accordingly, Bentz *et al*. [36] show that there is a strong correlation between an information-theoretical measure that estimates the amount of redundancy contributed by the within-word structure [37] and the *WALS* index of morphological complexity. The question whether and how linguistic redundancy translates into descriptional redundancy (i.e. entropy) is an important avenue for future research.

At the same time, however, and in correspondence with the results of Lupyan & Dale [1, §11], the strongest and most unequivocal association was obtained between speaker population size and entropy rates: languages with more speakers tend to have higher entropy rates. What could potentially generate this result? Let me speculate and make two assumptions: (i) the language learners of the current generation are the corpus-generators for future generations of language learners [55], (ii) each speaker has a slightly different mental representation of the statistical structure of language, or put differently, each speaker has her 'own characteristic entropy' [56]. In combination, (i) and (ii) in conjunction with classic population-based models of cultural transmission [57] potentially imply that a modified version of the mechanism discussed by Nettle [2] might work: the smaller the speech community, the higher the 'degree of expected overlap' in the sample of utterances different learners are exposed to when they learn the language. So, the idea here is that in languages with only few speakers, the current generation learns the language from a smaller set of individuals, who have had a greater chance to interact with each other and have a greater chance of having learned the language from the same individuals, compared to speech communities with many members. In the latter case, language learners will experience more variation in the input they receive. (It is worth mentioning in this context that language acquisition research indicates that exposure to more speakers facilitates acquisition at the phonological level, as one anonymous reviewer pointed out, e.g. [58] and the references therein.) This greater input variation, in turn, could lead to a gradual accumulation of variable statistical structure that could be reflected in more information-theoretic uncertainty (i.e. entropy) for languages with more speakers. One corollary of this mechanism would be that (the statistical structure of) languages with more speakers should, on average, be harder to learn. Further work is clearly needed to support such a 'population thinking' account [59].

Research ethics. Approval from a research ethics committee was not required, as this study involved no human or animal participants.

Data accessibility. Data and (Stata 14) code required to reproduce all results presented in this paper are available as electronic supplementary material.

Competing interests. I have no competing interests.

Funding. I received no specific funding for this work. But, the publication of this article was partially funded by the Open Access Fund of the Leibniz Association.

Acknowledgements. I thank Chris Bentz and Gary Lupyan for sharing data and their thoughts on the linguistic niche hypothesis with me. I also thank Marc Brysbaert, Kaius Sinnemäki, Sascha Wolfer and one anonymous reviewer for input and feedback and Sarah Signer for proofreading.

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
