## [Reviewer comments · Royal Society Open Science]

Review History

RSOS-181274.R0 (Original submission)

Review form: Reviewer 1 (Marc Brysbaert)

Is the manuscript scientifically sound in its present form?

Yes

Are the interpretations and conclusions justified by the results?

Yes

Is the language acceptable?

Yes

Is it clear how to access all supporting data?

Yes

Do you have any ethical concerns with this paper?

No

Have you any concerns about statistical analyses in this paper?

No

Recommendation?

Accept with minor revision (please list in comments)

Comments to the Author(s)

Review ms Koplénig

Marc Brysbaert

I must admit I like the ms, even though I thought the niche hypothesis had good credibility (it has been argued that in Dutch too the language simplified at times of mass immigration). This is surely going to rock the boat. However, as far as I can judge, the author is completely transparent about the variables used (mainly coming from other sources) and the analyses done.

My main concern was that the niche hypothesis not really requires control for population size. It is just the proportion of non-native adult speakers that matters. So, I am not 100% convinced the present analyses will be enough for everyone (although it is true that the effect does not survive control for other random factors either and that the effect of population size remains significant when L2 proportion is partialled out). I was reminded of a recent article in which the conclusion was that for this type of correlational questions the conclusions largely depend on the statistics used and the variables taken into account or not (see <http://journals.sagepub.com/doi/10.1177/2515245917747646>). So, different authors may come to different conclusions.

The following were minor questions I had:

Lines 93-94 (and further on again): Does this mean that 3 and 4 were scored double? Why not 3 and lower and 4 and higher, or lower than 4 and higher than 3?

Line 120: Are you sure about this equation for population size? It seems to indicate that the parameter increases the higher estimated L2 proportion is (going to infinity for a language in which everyone is L2 speaker; e.g., Esperanto).

It is always good to keep in mind that zero correlations can also be caused by unreliable variables. If other people were to compile the same measures, how much would they correlate with the present ones? To some extent, I am satisfied to see that all measures have been collected from good sources (and are made available), but still it is something to keep in mind, certainly because we are talking about quite a few understudied languages here.

Review form: Reviewer 2 (Vera Kempe)**Is the manuscript scientifically sound in its present form?**

No

Are the interpretations and conclusions justified by the results?

Yes

Is the language acceptable?

Yes

Is it clear how to access all supporting data?

Yes

Do you have any ethical concerns with this paper?

No

Have you any concerns about statistical analyses in this paper?

Yes

Recommendation?

Accept with minor revision (please list in comments)

Comments to the Author(s)

The paper tests the hypothesized link between proportion of L2-speakers and morphological complexity for a significantly larger sample of languages than has been attempted previously. The findings show that when population size is partialled out then neither vehicularity (an indicator as to whether a language is used by L2-speakers) nor proportion of L2-speakers show an independent association with morphological complexity or with information-theoretical entropy. Instead, there is an indication that population size is associated with lower morphological complexity and higher entropy pointing towards explanations of change in linguistic structure to do with the size and structure of social networks.

Having previously expressed doubts myself that there should be a plausible link between proportion of L2-speakers and morphological complexity I am very sympathetic to the overall aims of this paper. I therefore would like my comments to be viewed as suggestions for improvement before publication.

My main concern is that in order to make a strong contribution I find that the paper needs to improve in clarity to appeal to a broader audience beyond computational linguistics.

Specifically, at the beginning of the Methods Section, I suggest a clearer roadmap for the analyses. There is some redundancy in the explanation of the permutation process at the expense of clarity with respect to the overall logic of the analysis. While reading through it the first time, it was not always clear to me whether the paper focuses on testing the relationship between proportion of L2-speakers and morphological complexity (as the title suggests), the relationship between total number of speakers and proportion of L2-speakers, the relationship between total number of speakers and morphological complexity or all three of them. More clarity about the specific aims is needed. In particular, the author has opted for using partial correlations but having established a link between population size and morphological complexity/entropy I was wondering whether a mediation analysis (Hayes, 2009) is a more appropriate and comprehensive way of testing what the Linguistic Niche Hypothesis asserts -- that the link between population size and morphological complexity is mediated by proportion of L2-speakers.

Hayes, A. F. (2009). Beyond Baron and Kenny: Statistical mediation analysis in the new millennium. *Communication monographs*, 76(4), 408-420.

In addition, it would be desirable to provide a clear summary of how the various constructs (proportion of L2-speakers, morphological complexity and information-theoretic complexity/entropy) are operationalized computationally, and what the rationale is. This needs to be explained in the text so readers can appreciate whether the measures chosen are good measures. Especially in light of the result showing a link between population size and entropy, and the associated discussion, a clearer explanation in the Methods section of how entropy is

calculated would be necessary, to understand the links between morphological complexity and redundancy (e.g. intra-word vs. between words). For example, how would redundant gender marking in determiner-adjective-noun agreement patterns (e.g. German: eine warme Suppe) be accounted for in the entropy measures?

Below, I am listing some more specific concerns and suggestions (numbers = lines):

97: "In this paper, I use available information on both L2 users and on vehicularity to test the assumed relationship between the proportion of adult speakers and both morphological and information-theoretic complexity..." -- At this point, the author should provide a clear hypothesis as to what to expect if the formulation of the LNH that attributes a link between population size and morphological complexity to the proportion of L2-speakers is true. I would have expected something along the lines of 'According to the Linguistic Niche Hypothesis, languages with an EGIDS > 3 (or with vehicularity = 1) should have higher entropy/lower morphological complexity when statistically controlling for population size'.

104: The author introduces measures of geographical language area yet the theoretical relevance of this variable has not been explained and it does not seem to figure as a random effect in the analyses.

115: The statement "Language with an EGIDS value of 3 and higher are categorised as vehicular, while languages with an EGIDS of 4 or lower are categorized as non-vehicular." leaves it unclear as to how languages with an EGIDS of 3 are categorised. I suspect it may be the other way around: 3 and lower are vehicular? Also, correct 'Language' to 'Languages'.

145: 'The median estimated speaker population size is 15,000; the median estimated proportion of L2 speakers is 0 for non-vehicular languages and 0.233 for vehicular languages. Moreover, the statement in line 151 suggests 'According to the definition of the Ethnologue, a vehicular language should have a significant number of L2 users, while no L2 users are expected for non-vehicular languages.' Contradicting these two statements, the data file in the Supplementary Materials shows that some non-vehicular languages are listed with a certain proportion of L2 speakers. It is thus not clear as to what exactly proportion of L2-speakers measures. For example, the proportion of 0.62 of L2 speakers for English with a vehicularity index of 1 seems to suggest that 62% of English speakers use it as an L2 (or L3) in addition to their L1. But then the data file also lists an L2 proportion of 0.26 for Balkan Romani, which has a vehicularity index of 0. This suggests that 26% of speakers for whom Romani is the L1 also know another L2, given that Romani is not categorised as a vehicular language. The paper seems to treat these two situations as equivalent: if a language is being spoken by a lot of non-native speakers as in the case of English this is apparently treated in the same way as a native language some speakers of which also use another L2. If there is mutual influence in both cases then the underlying cognitive mechanisms must be fundamentally different: accommodation to non-native grammar learning difficulties in the former case vs. language attrition in the latter. I have my strong reservations as to whether it is correct to equate those two very different situations. In any case, the relationship between the two variables and the fact that not all languages with a vehicularity of 0 have 0 L2 speakers needs to be dealt with openly. I wonder whether it is an artefact of the way proportion of L2-speakers is calculated in Ethnologue.

As indicated above, the paragraph between lines 181 and 200 is rather redundant as it seems to explain the rationale behind the permutation process again – this could be summarised more succinctly as pertaining to both sets of analyses.

183: Explain what is meant by 'functional form' of a variable.

211: Explain the rationale behind conducting analyses for 6 language families and language areas.

254/255: The sentence should be edited to refer to information-theoretical and not morphological complexity.

267-270: Please rephrase as the findings do not contradict an influence of the environment per se, as the link between population size and entropy/morphological complexity indicates; they just do not support the idea that this influence is due to the proportion of L2 speakers.

Page 19, line 48: Table 2 is mislabelled as Table 1.

Decision letter (RSOS-181274.R0)

08-Nov-2018

Dear Dr Koplenig,

The editors assigned to your paper ("Language structure does not seem to be influenced by the proportion of non-native speakers") have now received comments from reviewers. We would like you to revise your paper in accordance with the referee and Associate Editor suggestions which can be found below (not including confidential reports to the Editor). Please note this decision does not guarantee eventual acceptance.

Please submit a copy of your revised paper before 01-Dec-2018. Please note that the revision deadline will expire at 00.00am on this date. If we do not hear from you within this time then it will be assumed that the paper has been withdrawn. In exceptional circumstances, extensions may be possible if agreed with the Editorial Office in advance. We do not allow multiple rounds of revision so we urge you to make every effort to fully address all of the comments at this stage. If deemed necessary by the Editors, your manuscript will be sent back to one or more of the original reviewers for assessment. If the original reviewers are not available, we may invite new reviewers.

If your study uses humans or animals please include details of the ethical approval received, including the name of the committee that granted approval. For human studies please also detail

whether informed consent was obtained. For field studies on animals please include details of all permissions, licences and/or approvals granted to carry out the fieldwork.

- Data accessibility

If you wish to submit your supporting data or code to Dryad (<http://datadryad.org/>), or modify your current submission to dryad, please use the following link:
<http://datadryad.org/submit?journalID=RSOS&manu=RSOS-181274>

- Competing interests

- Authors' contributions

- Acknowledgements

- Funding statement

Please note that Royal Society Open Science charge article processing charges for all new submissions that are accepted for publication. Charges will also apply to papers transferred to Royal Society Open Science from other Royal Society Publishing journals, as well as papers submitted as part of our collaboration with the Royal Society of Chemistry (<http://rsos.royalsocietypublishing.org/chemistry>). If your manuscript is newly submitted and subsequently accepted for publication, you will be asked to pay the article processing charge, unless you request a waiver and this is approved by Royal Society Publishing. You can find out

more about the charges at <http://rsos.royalsocietypublishing.org/page/charges>. Should you have any queries, please contact openscience@royalsociety.org.

on behalf of Prof Antonia Hamilton (Subject Editor)
openscience@royalsociety.org

Associate Editor's comments:

Thank you for this submission. As you will see, the reviewers are broadly in favour of eventual publication of your paper, but each recommend improvements that we would like you to incorporate into your work (or at the very least provide a reasoned rebuttal as to why you have opted not to do so). Good luck!

Comments to Author:

Reviewers' Comments to Author:
Reviewer: 1

Comments to the Author(s)
Review ms Kopleinig

Marc Brysbaert

I must admit I like the ms, even though I thought the niche hypothesis had good credibility (it has been argued that in Dutch too the language simplified at times of mass immigration). This is surely going to rock the boat. However, as far as I can judge, the author is completely transparent about the variables used (mainly coming from other sources) and the analyses done.

My main concern was that the niche hypothesis not really requires control for population size. It is just the proportion of non-native adult speakers that matters. So, I am not 100% convinced the present analyses will be enough for everyone (although it is true that the effect does not survive control for other random factors either and that the effect of population size remains significant when L2 proportion is partialled out). I was reminded of a recent article in which the conclusion was that for this type of correlational questions the conclusions largely depend on the statistics used and the variables taken into account or not (see <http://journals.sagepub.com/doi/10.1177/2515245917747646>). So, different authors may come to different conclusions.

The following were minor questions I had:

Lines 93-94 (and further on again): Does this mean that 3 and 4 were scored double? Why not 3 and lower and 4 and higher, or lower than 4 and higher than 3?

Line 120: Are you sure about this equation for population size? It seems to indicate that the parameter increases the higher estimated L2 proportion is (going to infinity for a language in which everyone is L2 speaker; e.g., Esperanto).

It is always good to keep in mind that zero correlations can also be caused by unreliable variables. If other people were to compile the same measures, how much would they correlate with the present ones? To some extent, I am satisfied to see that all measures have been collected from good sources (and are made available), but still it is something to keep in mind, certainly because we are talking about quite a few understudied languages here.

Reviewer: 2

Comments to the Author(s)

The paper tests the hypothesized link between proportion of L2-speakers and morphological complexity for a significantly larger sample of languages than has been attempted previously. The findings show that when population size is partialled out then neither vehicularity (an indicator as to whether a language is used by L2-speakers) nor proportion of L2-speakers show an independent association with morphological complexity or with information-theoretical entropy. Instead, there is an indication that population size is associated with lower morphological complexity and higher entropy pointing towards explanations of change in linguistic structure to do with the size and structure of social networks.

Having previously expressed doubts myself that there should be a plausible link between proportion of L2-speakers and morphological complexity I am very sympathetic to the overall aims of this paper. I therefore would like my comments to be viewed as suggestions for improvement before publication.

My main concern is that in order to make a strong contribution I find that the paper needs to improve in clarity to appeal to a broader audience beyond computational linguistics.

Specifically, at the beginning of the Methods Section, I suggest a clearer roadmap for the analyses. There is some redundancy in the explanation of the permutation process at the expense of clarity with respect to the overall logic of the analysis. While reading through it the first time, it was not always clear to me whether the paper focuses on testing the relationship between proportion of L2-speakers and morphological complexity (as the title suggests), the relationship between total number of speakers and proportion of L2-speakers, the relationship between total number of speakers and morphological complexity or all three of them. More clarity about the specific aims is needed. In particular, the author has opted for using partial correlations but having established a link between population size and morphological complexity/entropy I was wondering whether a mediation analysis (Hayes, 2009) is a more appropriate and comprehensive way of testing what the Linguistic Niche Hypothesis asserts -- that the link between population size and morphological complexity is mediated by proportion of L2-speakers.

Hayes, A. F. (2009). Beyond Baron and Kenny: Statistical mediation analysis in the new millennium. *Communication monographs*, 76(4), 408-420.

In addition, it would be desirable to provide a clear summary of how the various constructs (proportion of L2-speakers, morphological complexity and information-theoretic complexity/entropy) are operationalized computationally, and what the rationale is. This needs to be explained in the text so readers can appreciate whether the measures chosen are good

measures. Especially in light of the result showing a link between population size and entropy, and the associated discussion, a clearer explanation in the Methods section of how entropy is calculated would be necessary, to understand the links between morphological complexity and redundancy (e.g. intra-word vs. between words). For example, how would redundant gender marking in determiner-adjective-noun agreement patterns (e.g. German: eine warme Suppe) be accounted for in the entropy measures?

Below, I am listing some more specific concerns and suggestions (numbers = lines):

97: "In this paper, I use available information on both L2 users and on vehicularity to test the assumed relationship between the proportion of adult speakers and both morphological and information-theoretic complexity..." -- At this point, the author should provide a clear hypothesis as to what to expect if the formulation of the LNH that attributes a link between population size and morphological complexity to the proportion of L2-speakers is true. I would have expected something along the lines of 'According to the Linguistic Niche Hypothesis, languages with an EGIDS > 3 (or with vehicularity = 1) should have higher entropy/lower morphological complexity when statistically controlling for population size'.

104: The author introduces measures of geographical language area yet the theoretical relevance of this variable has not been explained and it does not seem to figure as a random effect in the analyses.

115: The statement "Language with an EGIDS value of 3 and higher are categorised as vehicular, while languages with an EGIDS of 4 or lower are categorized as non-vehicular." leaves it unclear as to how languages with an EGIDS of 3 are categorised. I suspect it may be the other way around: 3 and lower are vehicular? Also, correct 'Language' to 'Languages'.

145: 'The median estimated speaker population size is 15,000; the median estimated proportion of L2 speakers is 0 for non-vehicular languages and 0.233 for vehicular languages. Moreover, the statement in line 151 suggests 'According to the definition of the Ethnologue, a vehicular language should have a significant number of L2 users, while no L2 users are expected for non-vehicular languages.' Contradicting these two statements, the data file in the Supplementary Materials shows that some non-vehicular languages are listed with a certain proportion of L2 speakers. It is thus not clear as to what exactly proportion of L2-speakers measures. For example, the proportion of 0.62 of L2 speakers for English with a vehicularity index of 1 seems to suggest that 62% of English speakers use it as an L2 (or L3) in addition to their L1. But then the data file also lists an L2 proportion of 0.26 for Balkan Romani, which has a vehicularity index of 0. This suggests that 26% of speakers for whom Romani is the L1 also know another L2, given that Romani is not categorised as a vehicular language. The paper seems to treat these two situations as equivalent: if a language is being spoken by a lot of non-native speakers as in the case of English this is apparently treated in the same way as a native language some speakers of which also use another L2. If there is mutual influence in both cases then the underlying cognitive mechanisms must be fundamentally different: accommodation to non-native grammar learning difficulties in the former case vs. language attrition in the latter. I have my strong reservations as to whether it is correct to equate those two very different situations. In any case, the relationship between the two variables and the fact that not all languages with a vehicularity of 0 have 0 L2 speakers needs to be dealt with openly. I wonder whether it is an artefact of the way proportion of L2-speakers is calculated in Ethnologue.

As indicated above, the paragraph between lines 181 and 200 is rather redundant as it seems to explain the rationale behind the permutation process again – this could be summarised more succinctly as pertaining to both sets of analyses.

183: Explain what is meant by 'functional form' of a variable.

211: Explain the rationale behind conducting analyses for 6 language families and language areas.

254/255: The sentence should be edited to refer to information-theoretical and not morphological complexity.

267-270: Please rephrase as the findings do not contradict an influence of the environment per se, as the link between population size and entropy/morphological complexity indicates; they just do not support the idea that this influence is due to the proportion of L2 speakers.

Page 19, line 48: Table 2 is mislabelled as Table 1.

Author's Response to Decision Letter for (RSOS-181274.R0)

See Appendix A.

RSOS-181274.R1 (Revision)

Review form: Reviewer 1 (Marc Brysbaert)

Is the manuscript scientifically sound in its present form?

Yes

Are the interpretations and conclusions justified by the results?

Yes

Is the language acceptable?

Yes

Is it clear how to access all supporting data?

Yes

Do you have any ethical concerns with this paper?

No

Have you any concerns about statistical analyses in this paper?

No

Recommendation?

Accept as is

Comments to the Author(s)

Review revision Koplenig

Marc Brysbaert

This is a revision of a ms I reviewed favorably before. Given that the author has responded well to the suggestions made, I am happy to recommend the paper for publication now.

I foresee that the present analyses will be called into question and that alternative analyses will provide some (small) support for the linguistic niche hypothesis. However, this is a discussion that must take place openly in the literature and not as part of hidden reviews. Kopleinig has a strong case to question the niche hypothesis and now it is up to the proponents to show him and the readers what they have misunderstood. If the proponents are not able to do so convincingly, Kopleinig's alternative hypothesis can be the start of further progress.

Review form: Reviewer 2 (Vera Kempe)

Is the manuscript scientifically sound in its present form?

Yes

Are the interpretations and conclusions justified by the results?

Yes

Is the language acceptable?

Yes

Is it clear how to access all supporting data?

Yes

Do you have any ethical concerns with this paper?

No

Have you any concerns about statistical analyses in this paper?

No

Recommendation?

Accept with minor revision (please list in comments)

Comments to the Author(s)

The author did an excellent job addressing the concerns I expressed in my previous review, including providing a clear statement of hypotheses. This is very helpful.

I particularly welcome the additional analysis that remove the languages with a vehicularity index of 0 and a proportion of L2-speakers > 0 . However, I think languages that lack the estimate of L2 proportion should also be removed. I am sorry I am being very nit-picky about this but I am still not convinced that vehicularity is a good proxy for proportion of adult L2-speakers of a language. The author states that where there was no estimated proportion of L2 speakers available an estimated proportion of 0 was entered but this may not be a good way of dealing with this lack of information. Take Scottish Gaelic. It is listed with a vehicularity index of 1 and misses the estimate of L2 speakers. Presumably a 0 was entered here. However, this would mischaracterise the situation that there are no monolingual Gaelic speakers in Scotland and that one would be hard-pressed to argue that all Gaelic speakers use the language as an L2. Gaelic is a

clear case of communal bilingualism where the language is acquired by children alongside English, not just in the home but increasingly in Gaelic-medium schools. How many other languages are there in the relatively small set of 241 vehicular languages where it is not clear what the L1 and what the L2 is? I am presenting this example only to express my doubts that vehicularity is a good estimate for the role of adult L2-learners. I think the assumption that inter-generational disruption as measured by the EGIDS is linked to use as an L2 is probably not correct. If anything, I would think that languages with high EGIDS values may also be heritage languages, i.e. languages learned incompletely by children. While this also may affect language structure in interesting ways it is different from the assumption that it is the learning constraints of adults that influence these languages, as stated by the LNH.

Because I think that the point the author makes is a very important one I want to make sure the arguments and the underpinning statistics are compelling. I would therefore suggest presenting the analysis that removes the 78 languages with a vehicularity index of 0 and a proportion of L2-speakers > 0 , and also exclude those languages that have no estimated proportion of L2-users (like Scottish Gaelic), and present this more conservative analysis in the main paper rather than in the Supplementary materials.

This then raises another issue: When those 78 languages with ambiguous vehicularity are removed both DVs show a weak but significant relationship with proportion of L2-speakers. I think it is therefore not very convincing to entirely reject the idea that L2-speakers are a driving force behind morphological simplification in languages with more speakers. I suggest rephrasing the discussion a bit more cautiously by saying that proportion of L2-speakers may only play a minor role compared to other factors associated with population size, rather than stating that it plays no role altogether. Because I think the point the author makes is a very important one I am just encouraging him to be as cautious and conservative as possible, to convince those who attribute (undue -- in my view) importance to adult L2-learning as a driving force.

While the quantification of entropy has been made much clearer, I think some readers may still be confused about the quantification of morphological complexity. Especially the notion of 'feature value' is not immediately clear without further explanation. I had to consult the Bentz et al. (2016) Proceedings paper to fully understand what is meant by this. Providing an example, e.g. with respect to gender, might be helpful here. I also think that some more critical comments about this particular way of quantifying complexity may be in order, especially when it comes to explaining the lack of the expected correlation between entropy and morphological complexity: One reason may be that feature value is not a good, or should I say, sufficient, way of quantifying complexity. Take gender again: In languages where gender is marked by systematic morphological cues, e.g. Russian or Spanish, within-word entropy should be smaller than in gender languages where there is a lot of arbitrariness in the gender category assignment, like German. I understand this is a more substantial debate to be had and the author alludes to that but I think readers may expect a slightly more thorough discussion of this issue – perhaps add one or two sentences about the pros and cons of quantification of morphological complexity in this way and how this may relate to entropy, in addition to the interesting comments about within- and between-word entropy.

The suggestion that greater variability associated with larger population sizes may contribute to higher entropy is an interesting one but should be reconciled with the literature on phonological category learning which has demonstrated that being exposed to more speakers facilitates learning as it allows learners to distinguish indexical from category-relevant acoustic features. Again, this is a big and interesting debate to be had but a reference to this literature may be in order.

Minor points:

The abstract refers to 'adult speakers'. This is misleading as any native speaker will eventually end up being an adult. I think you mean 'adult learners' and I would stick with this label.

Page 11, line 59: The sentence that starts with 'But also relevant...' is incomplete - please rephrase.

Decision letter (RSOS-181274.R1)

09-Jan-2019

Dear Dr Koplenig:

Manuscript ID RSOS-181274.R1 entitled "Language structure is influenced by the number of speakers but seemingly not by the proportion of non-native speakers" which you submitted to Royal Society Open Science, has been reviewed. The comments of the reviewer(s) are included at the bottom of this letter.

Please submit a copy of your revised paper before 01-Feb-2019. Please note that the revision deadline will expire at 00.00am on this date. If we do not hear from you within this time then it will be assumed that the paper has been withdrawn. In exceptional circumstances, extensions may be possible if agreed with the Editorial Office in advance. We do not allow multiple rounds of revision so we urge you to make every effort to fully address all of the comments at this stage. If deemed necessary by the Editors, your manuscript will be sent back to one or more of the original reviewers for assessment. If the original reviewers are not available we may invite new reviewers.

- Ethics statement

- Data accessibility

- Competing interests

- Authors' contributions

- Acknowledgements

- Funding statement

Kind regards,

Royal Society Open Science Editorial Office
Royal Society Open Science
openscience@royalsociety.org

on behalf of Professor Antonia Hamilton (Subject Editor)
openscience@royalsociety.org

Associate Editor Comments to Author:

As you'll see the referees are generally enthusiastic about both your paper and the revisions included. That being said, we'd like you to look again at Reviewer 2's suggestions -- not because the paper is not publishable as is but because that reviewer thinks that addressing their points will substantially add to the success of your paper, when published. We'd encourage you to take their points into consideration when submitting the revision.

Reviewer comments to Author:

Reviewer: 2

Comments to the Author(s)

The author did an excellent job addressing the concerns I expressed in my previous review, including providing a clear statement of hypotheses. This is very helpful.

I particularly welcome the additional analysis that remove the languages with a vehicularity index of 0 and a proportion of L2-speakers > 0 . However, I think languages that lack the estimate of L2 proportion should also be removed. I am sorry I am being very nit-picky about this but I am still not convinced that vehicularity is a good proxy for proportion of adult L2-speakers of a language. The author states that where there was no estimated proportion of L2 speakers available an estimated proportion of 0 was entered but this may not be a good way of dealing with this lack of information. Take Scottish Gaelic. It is listed with a vehicularity index of 1 and misses the estimate of L2 speakers. Presumably a 0 was entered here. However, this would mischaracterise the situation that there are no monolingual Gaelic speakers in Scotland and that one would be hard-pressed to argue that all Gaelic speakers use the language as an L2. Gaelic is a clear case of communal bilingualism where the language is acquired by children alongside English, not just in the home but increasingly in Gaelic-medium schools. How many other languages are there in the relatively small set of 241 vehicular languages where it is not clear what the L1 and what the L2 is? I am presenting this example only to express my doubts that vehicularity is a good estimate for the role of adult L2-learners. I think the assumption that inter-generational disruption as measured by the EGIDS is linked to use as an L2 is probably not correct. If anything, I would think that languages with high EGIDS values may also be heritage languages, i.e. languages learned incompletely by children. While this also may affect language structure in interesting ways it is different from the assumption that it is the learning constraints of adults that influence these languages, as stated by the LNH.

Because I think that the point the author makes is a very important one I want to make sure the arguments and the underpinning statistics are compelling. I would therefore suggest presenting the analysis that removes the 78 languages with a vehicularity index of 0 and a proportion of L2-speakers > 0 , and also exclude those languages that have no estimated proportion of L2-users (like Scottish Gaelic), and present this more conservative analysis in the main paper rather than in the Supplementary materials.

This then raises another issue: When those 78 languages with ambiguous vehicularity are removed both DVs show a weak but significant relationship with proportion of L2-speakers. I think it is therefore not very convincing to entirely reject the idea that L2-speakers are a driving force behind morphological simplification in languages with more speakers. I suggest rephrasing the discussion a bit more cautiously by saying that proportion of L2-speakers may only play a minor role compared to other factors associated with population size, rather than stating that it plays no role altogether. Because I think the point the author makes is a very important one I am just encouraging him to be as cautious and conservative as possible, to convince those who attribute (undue -- in my view) importance to adult L2-learning as a driving force.

While the quantification of entropy has been made much clearer, I think some readers may still be confused about the quantification of morphological complexity. Especially the notion of 'feature value' is not immediately clear without further explanation. I had to consult the Bentz et al. (2016) Proceedings paper to fully understand what is meant by this. Providing an example, e.g. with respect to gender, might be helpful here. I also think that some more critical comments about this particular way of quantifying complexity may be in order, especially when it comes to explaining the lack of the expected correlation between entropy and morphological complexity: One reason may be that feature value is not a good, or should I say, sufficient, way of quantifying complexity. Take gender again: In languages where gender is marked by systematic morphological cues, e.g. Russian or Spanish, within-word entropy should be smaller than in gender languages where there is a lot of arbitrariness in the gender category assignment, like German. I understand this is a more substantial debate to be had and the author alludes to that but I think readers may expect a slightly more thorough discussion of this issue – perhaps add one or two sentences about the pros and cons of quantification of morphological complexity in this way and how this may relate to entropy, in addition to the interesting comments about within- and between-word entropy.

The suggestion that greater variability associated with larger population sizes may contribute to higher entropy is an interesting one but should be reconciled with the literature on phonological category learning which has demonstrated that being exposed to more speakers facilitates learning as it allows learners to distinguish indexical from category-relevant acoustic features. Again, this is a big and interesting debate to be had but a reference to this literature may be in order.

Minor points:

The abstract refers to 'adult speakers'. This is misleading as any native speaker will eventually end up being an adult. I think you mean 'adult learners' and I would stick with this label.

Page 11, line 59: The sentence that starts with 'But also relevant...' is incomplete – please rephrase.

Reviewer: 1

Comments to the Author(s)
Review revision Koplenig

Marc Brysbaert

This is a revision of a ms I reviewed favorably before. Given that the author has responded well to the suggestions made, I am happy to recommend the paper for publication now.

I foresee that the present analyses will be called into question and that alternative analyses will provide some (small) support for the linguistic niche hypothesis. However, this is a discussion that must take place openly in the literature and not as part of hidden reviews. Koplenig has a strong case to question the niche hypothesis and now it is up to the proponents to show him and the readers what they have misunderstood. If the proponents are not able to do so convincingly, Koplenig's alternative hypothesis can be the start of further progress.

Author's Response to Decision Letter for (RSOS-181274.R1)

See Appendix B.

RSOS-181274.R2 (Revision)

Review form: Reviewer 2 (Vera Kempe)

Is the manuscript scientifically sound in its present form?

Yes

Are the interpretations and conclusions justified by the results?

Yes

Is the language acceptable?

Yes

Is it clear how to access all supporting data?

Yes

Do you have any ethical concerns with this paper?

No

Have you any concerns about statistical analyses in this paper?

No

Recommendation?

Accept as is

Comments to the Author(s)

While I would still maintain that Scottish Gaelic should not be classified as a vehicular language as it is spoken in families and communities in the West of Scotland and only select few learn it as adults I am satisfied that the additional analysis the author has undertaken provide all the necessary statistical safeguards and that the Discussion is now phrased sufficiently cautiously, and I therefore look forward to seeing this very important paper in print.

Decision letter (RSOS-181274.R2)

29-Jan-2019

Dear Dr Kopleinig,

I am pleased to inform you that your manuscript entitled "Language structure is influenced by the number of speakers but seemingly not by the proportion of non-native speakers" is now accepted for publication in Royal Society Open Science.

on behalf of Dr Antonia Hamilton (Subject Editor)
openscience@royalsociety.org

Reviewer comments to Author:
Reviewer: 2

Comments to the Author(s)

While I would still maintain that Scottish Gaelic should not be classified as a vehicular language as it is spoken in families and communities in the West of Scotland and only select few learn it as adults I am satisfied that the additional analysis the author has undertaken provide all the necessary statistical safeguards and that the Discussion is now phrased sufficiently cautiously, and I therefore look forward to seeing this very important paper in print.

Follow Royal Society Publishing on Twitter: [@RSocPublishing](https://twitter.com/RSocPublishing)
Follow Royal Society Publishing on Facebook:
<https://www.facebook.com/RoyalSocietyPublishing.FanPage/>
Read Royal Society Publishing's blog: <https://blogs.royalsociety.org/publishing/>

Appendix A

Dear Editors,

First of all, I would like to thank both reviewers for their insightful comments. I think that the revised version of the paper benefited substantially from this input. I have carried out a number of revisions and added five new sections to the electronic supplementary material in order to improve the paper. Please find below find a detailed point-to-point reply [reviewer comments in italics, my responses in regular typeface].

Regards,

the author

Reviewer: 1

Review ms Koplénig

Marc Brysbaert

I must admit I like the ms, even though I thought the niche hypothesis had good credibility (it has been argued that in Dutch too the language simplified at times of mass immigration). This is surely going to rock the boat. However, as far as I can judge, the author is completely transparent about the variables used (mainly coming from other sources) and the analyses done.

*Thank you very much, below are my comments.

My main concern was that the niche hypothesis not really requires control for population size. It is just the proportion of non-native adult speakers that matters. So, I am not 100% convinced the present analyses will be enough for everyone (although it is true that the effect does not survive control for other random factors either and that the effect of population size remains significant when L2 proportion is partialled out). I was reminded of a recent article in which the conclusion was that for this type of correlational questions the conclusions largely depend on the statistics used and the variables taken into account or not (see <http://journals.sagepub.com/doi/10.1177/2515245917747646>). So, different authors may come to different conclusions.

*I agree. I changed the abstract, the title and the Introduction, to better clarify the aims of the paper. In addition, I added three new sections to the electronic supplementary material in order to improve the manuscript in terms of the raised concern: Section 5 contains a validation of the permutation test that suggests that it is indeed population size that explains the apparent relationship between *vehicularity* and complexity (morphological/information-theoretic). Section 6 contains an alternative permutation test that supports the results presented in the main part of the paper. Section 9 presents the results of a mediation analysis that was suggested by the second reviewer, so in a certain sense, this simulates a “different author”. The results of this test are in line with the other results.

The following were minor questions I had:

Lines 93-94 (and further on again): Does this mean that 3 and 4 were scored double? Why not 3 and lower and 4 and higher, or lower than 4 and higher than 3?

*I agree that this is ambiguous and modified this paragraph accordingly. Languages with an EGIDS value of 1, 2 or 3 are categorized as *vehicular*, while languages with an EGIDS value of 4 to 10 are categorized as *non-vehicular*.

Line 120: Are you sure about this equation for population size? It seems to indicate that the parameter increases the higher estimated L2 proportion is (going to infinity for a language in which everyone is L2 speaker; e.g., Esperanto).

*I agree that this was wrong. I modified the equation accordingly. In the analysis of the paper, I use the *L1* estimate. I added a section in the electronic supplementary (§8) where the total number of speakers ($L1 + L2$) is used as a measure of population size (see below).

It is always good to keep in mind that zero correlations can also be caused by unreliable variables. If other people were to compile the same measures, how much would they correlate with the present ones? To some extent, I am satisfied to see that all measures have been collected from good sources (and are made available), but still it is something to keep in mind, certainly because we are talking about quite a few understudied languages here.

*I agree. That is one of the reasons why I present separate analyses regarding morphological complexity on the full dataset (at least one available feature) and on a subset of languages

where at least six features are available (50% of all languages have information on at least six features). I modified the corresponding paragraph accordingly. I also modified the Discussion section to highlight the need for a common analytical approach in this context.

In addition to the actual p-values, I now also present Bonferroni adjusted p-values to account for multiple testing. The procedure is described in the Material and Methods section. I also added two new sections to the electronic supplementary material: Section 8 uses the total number of speakers as the measure of population size instead of the number of *L1* speakers. Section 7 presents separate analyses where all *non-vehicular* languages that are listed by the *Ethnologue* as having a proportion of *L2* speakers > 0 are dropped. All results presented in §7 and §8 generally support the results presented in the main part of the paper.

Reviewer: 2

The paper tests the hypothesized link between proportion of L2-speakers and morphological complexity for a significantly larger sample of languages than has been attempted previously. The findings show that when population size is partialled out then neither vehicularity (an indicator as to whether a language is used by L2-speakers) nor proportion of L2-speakers show an independent association with morphological complexity or with information-theoretical entropy. Instead, there is an indication that population size is associated with lower morphological complexity and higher entropy pointing towards explanations of change in linguistic structure to do with the size and structure of social networks.

Having previously expressed doubts myself that there should be a plausible link between proportion of L2-speakers and morphological complexity I am very sympathetic to the overall aims of this paper. I therefore would like my comments to be viewed as suggestions for improvement before publication.

*Thank you very much, below are my comments.

My main concern is that in order to make a strong contribution I find that the paper needs to improve in clarity to appeal to a broader audience beyond computational linguistics.

Specifically, at the beginning of the Methods Section, I suggest a clearer roadmap for the analyses. There is some redundancy in the explanation of the permutation process at the

expense of clarity with respect to the overall logic of the analysis. While reading through it the first time, it was not always clear to me whether the paper focuses on testing the relationship between proportion of L2-speakers and morphological complexity (as the title suggests), the relationship between total number of speakers and proportion of L2-speakers, the relationship between total number of speakers and morphological complexity or all three of them. More clarity about the specific aims is needed.

*I agree, I modified the abstract, the title and the Introduction to better clarify the aims of the paper. The redundancy regarding the permutation test has now been reduced.

In particular, the author has opted for using partial correlations but having established a link between population size and morphological complexity/entropy I was wondering whether a mediation analysis (Hayes, 2009) is a more appropriate and comprehensive way of testing what the Linguistic Niche Hypothesis asserts -- that the link between population size and morphological complexity is mediated by proportion of L2-speakers.

Hayes, A. F. (2009). Beyond Baron and Kenny: Statistical mediation analysis in the new millennium. Communication monographs, 76(4), 408-420.

*I agree that this is another interesting idea to interpret and test the Linguistic Niche Hypothesis. Accordingly, I added a new section to the electronic supplementary material (§9) that presents a binary mediation analysis. The results demonstrate that the indirect effect of *vehicularity* is not significantly different from zero in all investigated scenarios (this is checked via bootstrap confidence intervals with 10,000 repetitions each). This indicates that in essence the mediated proportion is meaningless and thus supports the results presented in the main part of the paper.

In addition, it would be desirable to provide a clear summary of how the various constructs (proportion of L2-speakers, morphological complexity and information-theoretic complexity/entropy) are operationalized computationally, and what the rationale is. This needs to be explained in the text so readers can appreciate whether the measures chosen are good measures. Especially in light of the result showing a link between population size and entropy, and the associated discussion, a clearer explanation in the Methods section of how entropy is calculated would be necessary, to understand the links between morphological complexity and redundancy (e.g. intra-word vs. between words). For example, how would

redundant gender marking in determiner-adjective-noun agreement patterns (e.g. German: eine warme Suppe) be accounted for in the entropy measures?

*I agree. I added an equation for all key quantities of interest. In addition, I added a whole new paragraph that explains how exactly the entropy rate is calculated and the underlying rationale. Regarding the question how linguistic redundancy translates into descriptive (i.e. information-theoretic) redundancy: I would like to remain rather agnostic here, because I believe that the results demonstrate that the relationship between both kinds of redundancy might not be as direct as previously asserted, for example by Lupyan and Dale (2010, Language Structure Is Partly Determined by Social Structure) or Nettle (2012, Social scale and structural complexity in human languages). I added a sentence to the Discussion section to highlight the need for additional research in this context.

Regarding the determiner-adjective-noun agreement example: from an information-theoretic point of view, the gender marking pattern could be understood in reference to data compression. Shannon showed that the minimal code length of an item with probability p is $\log_2(1/p)$. If we assume that somewhere in the text/corpus before the sentence of interest, there is a sentence like “eine warme Farbe”, then the compressor could use that information to compress/predict the example sentence, e.g. the conditional probability of “e” after “eine warm”, i.e. $p(\text{“e”}|\text{“eine warm”})$ will be very high, almost 1, so the code length will be almost $\log_2(1/1) \approx 0$. Therefore, the redundancy in gender marking can be used to compress the data.

Below, I am listing some more specific concerns and suggestions (numbers = lines):

97: *“In this paper, I use available information on both L2 users and on vehicularity to test the assumed relationship between the proportion of adult speakers and both morphological and information-theoretic complexity...” -- At this point, the author should provide a clear hypothesis as to what to expect if the formulation of the LNH that attributes a link between population size and morphological complexity to the proportion of L2-speakers is true. I would have expected something along the lines of ‘According to the Linguistic Niche Hypothesis, languages with an EGIDS > 3 (or with vehicularity = 1) should have higher entropy/lower morphological complexity when statistically controlling for population size’.*

*Agree, I have done so at the end of the Introduction.

104: *The author introduces measures of geographical language area yet the theoretical relevance of this variable has not been explained and it does not seem to figure as a random effect in the analyses.*

*As described at the beginning of the Results section, both speaker population size and geographical range size are used to test whether *vehicularity* can be used to indirectly test the inductive argument of the linguistic niche hypothesis. I have changed the Introduction to highlight this upfront.

115: *The statement “Language with an EGIDS value of 3 and higher are categorised as vehicular, while languages with an EGIDS of 4 or lower are categorized as non-vehicular.” leaves it unclear as to how languages with an EGIDS of 3 are categorised. I suspect it may be the other way around: 3 and lower are vehicular? Also, correct ‘Language’ to ‘Languages’.*

*I agree and modified this paragraph accordingly. Languages with an EGIDS value of 1,2 or 3 are categorized as vehicular, while languages with an EGIDS value of 4 to 10 are categorized as non-vehicular.

145: *‘The median estimated speaker population size is 15,000; the median estimated proportion of L2 speakers is 0 for non-vehicular languages and 0.233 for vehicular languages. Moreover, the statement in line 151 suggests ‘According to the definition of the Ethnologue, a vehicular language should have a significant number of L2 users, while no L2 users are expected for non-vehicular languages.’ Contradicting these two statements, the data file in the Supplementary Materials shows that some non-vehicular languages are listed with a certain proportion of L2 speakers. It is thus not clear as to what exactly proportion of L2-speakers measures. For example, the proportion of 0.62 of L2 speakers for English with a vehicularity index of 1 seems to suggest that 62% of English speakers use it as an L2 (or L3) in addition to their L1. But then the data file also lists an L2 proportion of 0.26 for Balkan Romani, which has a vehicularity index of 0. This suggests that 26% of speakers for whom Romani is the L1 also know another L2, given that Romani is not categorised as a vehicular language. The paper seems to treat these two situations as equivalent: if a language is being spoken by a lot of non-native speakers as in the case of English this is apparently treated in the same way as a native language some speakers of which also use another L2. If there is mutual influence in both cases then the underlying cognitive mechanisms must be*

fundamentally different: accommodation to non-native grammar learning difficulties in the former case vs. language attrition in the latter. I have my strong reservations as to whether it is correct to equate those two very different situations. In any case, the relationship between the two variables and the fact that not all languages with a vehicularity of 0 have 0 L2 speakers needs to be dealt with openly. I wonder whether it is an artefact of the way proportion of L2-speakers is calculated in Ethnologue.

* In the corresponding Materials and Methods section, I write that “*non-vehicular* languages with no available information on *L2* users are assigned an *L2* proportion of 0”. However, the reviewer is correct and I agree that I have not pointed this out clearly enough. I modified the section accordingly. *Ethnologue* asserts that for *non-vehicular* languages, “*L2* users are not expected”. However, there are in total 78 *non-vehicular* languages where *Ethnologue* reports an *L2* proportion that is greater than 0 (with a median estimate of 0.086). To rule out the possibility that those exceptions to the rule in the *Ethnologue* categorization scheme affect the results, separate analyses where those 78 languages are dropped are now presented in the electronic supplementary material (§7). In addition, the idea of using *vehicularity* as an indicator as to whether a language is used by *L2*-speakers in addition to the correlation analysis of *L2* proportion is also partly due to this.

As indicated above, the paragraph between lines 181 and 200 is rather redundant as it seems to explain the rationale behind the permutation process again – this could be summarised more succinctly as pertaining to both sets of analyses.

* I modified this accordingly.

183: Explain what is meant by ‘functional form’ of a variable.

*I modified this accordingly.

211: *Explain the rationale behind conducting analyses for 6 language families and language areas.*

*I agree. Section 4 of the electronic supplementary material now explains how the families/areas were selected.

254/255: The sentence should be edited to refer to information-theoretical and not morphological complexity.

*The sentence has been modified accordingly.

267-270: Please rephrase as the findings do not contradict an influence of the environment per se, as the link between population size and entropy/morphological complexity indicates; they just do not support the idea that this influence is due to the proportion of L2 speakers.

*Agree, the paragraph has been modified accordingly.

Page 19, line 48: Table 2 is mislabelled as Table 1.

* I agree, this typo has been changed.

Appendix B

Dear Editors,

First of all, I would like to thank both reviewers again for all their effort. I have carried out several revisions and added a new section to the electronic supplementary material in order to further improve the paper. Please find below find a detailed point-to-point reply [reviewer comments in italics, my responses in regular typeface].

Regards,

the author

Reviewer: 1

Review revision Koplénig

Marc Brysbaert

This is a revision of a ms I reviewed favorably before. Given that the author has responded well to the suggestions made, I am happy to recommend the paper for publication now.

I foresee that the present analyses will be called into question and that alternative analyses will provide some (small) support for the linguistic niche hypothesis. However, this is a discussion that must take place openly in the literature and not as part of hidden reviews.

Koplénig has a strong case to question the niche hypothesis and now it is up to the proponents to show him and the readers what they have misunderstood. If the proponents are not able to do so convincingly, Koplénig's alternative hypothesis can be the start of further progress.

*Thank you very much, I am looking forward to potential discussions. Since I have chosen the “open peer review” option when I submitted the paper, the peer review information (reviews and responses) will be published alongside the article, if it is accepted for publication.

Reviewer: 2

The author did an excellent job addressing the concerns I expressed in my previous review, including providing a clear statement of hypotheses. This is very helpful.

*Thank you very much, below are my comments.

I particularly welcome the additional analysis that remove the languages with a vehicularity index of 0 and a proportion of L2-speakers > 0 . However, I think languages that lack the estimate of L2 proportion should also be removed. I am sorry I am being very nit-picky about this but I am still not convinced that vehicularity is a good proxy for proportion of adult L2-speakers of a language.

*According to how the *Ethnologue* categorizes languages, *vehicularity* is a direct proxy for whether a language is used as an L2. As such it is treated in all analyses that use *vehicularity* as a binary predictor variable. Here is a corresponding interesting quote from the editors of the *Ethnologue* that shows how languages are classified: “Based on the use of the phrase “vehicular language” by some as a synonym for lingua franca, we use the term vehicular to refer to the extent to which a language is used to facilitate communication among those who speak different first languages. *If a language is characterized here as being Vehicular, it is used by others as an L2 in addition to being used by the community of L1 speakers.*” [1; my

emphasis]. In relation to the further comments below, I added a corresponding section to the Discussion section.

The author states that where there was no estimated proportion of L2 speakers available an estimated proportion of 0 was entered but this may not be a good way of dealing with this lack of information. Take Scottish Gaelic. It is listed with a vehicularity index of 1 and misses the estimate of L2 speakers. Presumably a 0 was entered here. However, this would mischaracterise the situation that there are no monolingual Gaelic speakers in Scotland and that one would be hard-pressed to argue that all Gaelic speakers use the language as an L2. Gaelic is a clear case of communal bilingualism where the language is acquired by children alongside English, not just in the home but increasingly in Gaelic-medium schools.

*As written in the main part of the paper and in correspondence with the categorization scheme of the Ethnologue [2], only non-vehicular languages with no available information on L2 users are assigned an L2 proportion of 0, because, as argued in the *Ethnologue* “such a language is by definition, a local language and L2 users are not expected” [2]. Scottish Gaelic is a good example here. It is categorized as being a *vehicular* language (EGIDS = 2). Since it misses the estimate of L2 speakers, it is only included in the *vehicularity* analyses, but it is not included in any of the correlation analyses. In the *vehicularity* analyses, it is treated as a language that is used as an L2.

How many other languages are there in the relatively small set of 241 vehicular languages where it is not clear what the L1 and what the L2 is? I am presenting this example only to express my doubts that vehicularity is a good estimate for the role of adult L2-learners. I think the assumption that inter-generational disruption as measured by the EGIDS is linked to use as an L2 is probably not correct. If anything, I would think that languages with high EGIDS values may also be heritage languages, i.e. languages learned incompletely by children. While this also may affect language structure in interesting ways it is different from

the assumption that it is the learning constraints of adults that influence these languages, as stated by the LNH.

*Of the 241 *vehicular* languages, 89 have information on the proportion of *L2* speakers, the remaining 152 languages are not used in any of the correlation analyses.

Because I think that the point the author makes is a very important one I want to make sure the arguments and the underpinning statistics are compelling. I would therefore suggest presenting the analysis that removes the 78 languages with a vehicularity index of 0 and a proportion of L2-speakers > 0, and also exclude those languages that have no estimated proportion of L2-users (like Scottish Gaelic), and present this more conservative analysis in the main paper rather than in the Supplementary materials.

*As written above, the analyses are already conservative. *Vehicular* languages that lack information on *L2* users are excluded from the correlation analyses. Only *non-vehicular* languages with no available *L2* information are assigned an *L2* proportion of 0. However, to fully convince the reviewer, I added a further section (10) to the electronic supplementary material. Here, I present additional analyses where only *vehicular* languages with available information regarding the proportion of *L2* speakers were used. I believe this to be a strong test for the linguistic niche hypothesis. As Table 12 demonstrates, this analysis does not lend support to the *LNH*, too. So, even for *vehicular* languages only, we cannot say that a high proportion of *L2* users is statistically associated with low morphological complexity and high entropy (when statistically controlling for the speaker population size).

This then raises another issue: When those 78 languages with ambiguous vehicularity are removed both DVs show a weak but significant relationship with proportion of L2-speakers. I think it is therefore not be very convincing to entirely reject the idea that L2-speakers are a driving force behind morphological simplification in languages with more speakers. I suggest

rephrasing the discussion a bit more cautiously by saying that proportion of L2-speakers may only play a minor role compared to other factors associated with population size, rather than stating that it plays no role altogether. Because I think the point the author makes is a very important one I am just encouraging him to be as cautious and conservative as possible, to convince those who attribute (undue -- in my view) importance to adult L2-learning as a driving force.

*Here, I am not sure what the reviewer means. Please note that Table 7 and Table 8 (Section 7) support the results presented in the main part of the paper. Compared to the main part of the paper, the only qualitative difference is row 2 of Table 8. It shows that there is a weak but significant negative correlation between morphological complexity and the L2 proportion. However, when controlling for the speaker population size, the correlation strength is sharply reduced (both absolute values are below 0.1) and only the correlation for the full dataset (at least one available *WALS* feature) is significant. In addition, both part correlation coefficients do not reach statistical significance after the *Bonferroni* adjustment. Apart from this point, there are no qualitative differences between Table 1 and Table 2 of the main part of the paper and Table 7 and Table 8. Thus, neither morphological complexity nor the entropy rate show a significant relationship with the L2 proportion as soon as the estimated speaker population size is controlled for.

I agree that it is a good idea to be *more cautiously* in the Discussion section, I updated it accordingly.

While the quantification of entropy has been made much clearer, I think some readers may still be confused about the quantification of morphological complexity. Especially the notion of 'feature value' is not immediately clear without further explanation. I had to consult the Bentz et al. (2016) Proceedings paper to fully understand what is meant by this. Providing an example, e.g. with respect to gender, might be helpful here.

*Agree, added the suggested example.

I also think that some more critical comments about this particular way of quantifying complexity may be in order, especially when it comes to explaining the lack of the expected correlation between entropy and morphological complexity: One reason may be that feature value is not a good, or should I say, sufficient, way of quantifying complexity. Take gender again: In languages where gender is marked by systematic morphological cues, e.g. Russian or Spanish, within-word entropy should be smaller than in gender languages where there is a lot of arbitrariness in the gender category assignment, like German. I understand this is a more substantial debate to be had and the author alludes to that but I think readers may expect a slightly more thorough discussion of this issue – perhaps add one or two sentences about the pros and cons of quantification of morphological complexity in this way and how this may relate to entropy, in addition to the interesting comments about within- and between-word entropy.

*Agree, I updated the Discussion section accordingly.

The suggestion that greater variability associated with larger population sizes may contribute to higher entropy is an interesting one but should be reconciled with the literature on phonological category learning which has demonstrated that being exposed to more speakers facilitates learning as it allows learners to distinguish indexical from category-relevant acoustic features. Again, this is a big and interesting debate to be had but a reference to this literature may be in order.

*Thank you for that, I added a corresponding remark and a literature reference.

Minor points:

The abstract refers to ‘adult speakers’. This is misleading as any native speaker will eventually end up being an adult. I think you mean ‘adult learners’ and I would stick with this label.

*Agree, I change that.

Page 11, line 59: The sentence that starts with ‘But also relevant...’ is incomplete – please rephrase.

*Agree, I changed that.

References

1. Lewis MP, Simons GF. 2010 Assessing Endangerment: Expanding Fishman’s GIDS. *Revue Roumaine de Linguistique* **55**, 103–120.
2. Simons GF, Fennig CD. 2017 Ethnologue Global Dataset, Twentieth edition. See <https://www.ethnologue.com/sites/default/files/Ethnologue-20-Global%20Dataset%20Doc.pdf>.